# Supplementation of Freezing Medium with Ginseng Improves Rooster Sperm Quality and Fertility Relative to Free Radicals and Antioxidant Enzymes

**DOI:** 10.3390/ani13162660

**Published:** 2023-08-18

**Authors:** Ruthaiporn Ratchamak, Supakorn Authaida, Thirawat Koedkanmark, Wuttigrai Boonkum, Yoswaris Semaming, Vibuntita Chankitisakul

**Affiliations:** 1Department of Animal Science, Faculty of Agricultural, Khon Kaen University, Khon Kaen 40002, Thailand; ruthaiporn.kung@gmail.com (R.R.); supakorn.u@kkumail.com (S.A.); thirawat.koe@kkumail.com (T.K.); wuttbo@kku.ac.th (W.B.); 2The Research and Development Network Center of Animal Breeding and Omics, Khon Kaen University, Khon Kaen 40002, Thailand; 3Program in Veterinary Technology, Faculty of Technology, Udon Thani Rajabhat University, Udon Thani 41000, Thailand; mai_jap@hotmail.com

**Keywords:** cryopreservation, fertility, lipid peroxidation, seminal plasma characteristics, Thai native chicken

## Abstract

**Simple Summary:**

Rooster sperm cryopreservation is the preferred technique for genetic preservation, particularly in indigenous breeds whose numbers are decreasing. However, cryopreserved semen is highly damaged, mainly from cold shock and oxidative stress during freezing, leading to membrane and sperm damage. It is necessary to protect sperm membranes against peroxidative damage to improve semen quality and fertility. This study aimed to determine the optimal ginseng concentration used as a freezing extender in cryopreserved Thai native rooster semen. Ginsenosides are active components extracted from ginseng that exert antioxidant effects that inhibit oxidative stress. Our results showed that ginseng extract dosage in the freezing extender influenced frozen-thawed semen quality, improved motility and membrane integrity, reduced oxidative stress caused by the freezing process, enhanced enzyme activity in seminal plasma, and increased fertility in rooster semen cryopreservation.

**Abstract:**

To the best of our knowledge, this study is the first to determine the effect of ginseng as an antioxidant supplement in freezing extenders on the quality of cryopreserved rooster semen. Semen samples were collected from 40 Thai native roosters (Pradu Hang Dum) using the dorso-abdominal massage method and then pooled and divided into five groups according to the concentrations of ginseng supplementation (0, 0.25, 0.50, 0.75, and 1 mg/mL) in a freezing extender. The semen suspensions were loaded into a medium straw and cryopreserved using the liquid nitrogen vapor method. The post-thaw semen was evaluated for sperm quality (sperm motility and membrane integrity), seminal plasma characteristics (lipid peroxidation, superoxide dismutase [SOD], catalase [CAT], and glutathione peroxidase [GPx]), and fertility. The results showed that ginseng extract supplementation at 0.25 mg/mL yielded the highest total motility, progressive motility, and membrane integrity (59.47%, 30.82%, and 48.30%, respectively; *p* < 0.05) in cryopreserved rooster semen. Higher malondialdehyde concentrations were observed in the control group than in the other groups (*p* < 0.05). SOD, CAT, and GPx increased compared with those in the control group (*p* < 0.05). The results showed that the fertility rate with 0.25 mg/mL of ginseng was higher than that of the control group (62.80% vs. 46.28%: *p* < 0.05). In conclusion, supplementation with 0.25 mg/mL of ginseng is recommended as an alternative component to the freezing extender to improve rooster semen cryopreservation.

## 1. Introduction

Traditionally, and in past decades, Thai native chickens were raised mainly for home consumption and played an important role in the economy at the community level [1]. However, they were raised on backyard farms by small farmers with poor husbandry practices, and insufficient nutrition resulted in poor growth performance [2], low egg production, and low fertility [3,4]. Therefore, they are now being replaced by commercial breeds or fast-growing broilers; thus, the number of native chickens has decreased seriously. Four Thai native chickens, including Pradu Hang Dam, have been promoted as a genetic resource for the purpose of sustainable utilization of indigenous chickens [5].

To conserve the genetic resources of those purebreds that have experienced a tremendous decline in their numbers and are considered endangered, sperm cryopreservation is an emerging potential tool for ex situ management of genetic diversity [6]. However, the success of this method in terms of freeze-thawed semen quality and fertility is highly variable in poultry because several kinds of cryo-damage occur in the sperm structure during cryopreservation [7,8]. These deleterious effects have been proposed to be associated with oxidative stress. Rooster semen cryopreservation significantly increases the concentration of malondialdehyde (MDA), the final product of lipid peroxidation [9]. Oxidation reactions occur at the plasma membrane, generating reactive oxygen species (ROS) such as superoxide (O^2−^), hydroxyl (OH), and hydrogen peroxide (H_2_O_2_) [10]. ROS have detrimental effects on sperm quality, particularly on movement and ATP synthesis discount [11]. The total and progressive motility percentage decrease varies from 30% to 50% [12], respectively, leading to the loss of fertility and of embryonic development after insemination with cryopreserved semen [13]. 

ROS generation during cryopreservation is the primary factor responsible for cryodamage. An effective antioxidant system that protects the sperm membranes against oxidative damage is essential. Partyka and Nizanski [14] reported that antioxidants that naturally contribute to chicken semen include superoxide dismutase (SOD), glutathione peroxidase (GPx), and catalase (CAT). These primary antioxidant enzymes protect sperm cells from oxidative damage by scavenging excess ROS during cryopreservation processing [15]. However, the levels of natural antioxidants are insufficient compared with the excessive production of ROS during preservation [16]. One possible strategy to detect damage from cryo-injury is to evaluate the effect of antioxidant supplementation in a semen extender. Antioxidants eliminate free radicals and convert their structures, thereby reducing the number of free radicals [17,18].

Ginseng is a traditional herb that is rich in bioactive phytochemicals. Ginsenosides are the predominant active components in ginseng extracts. It has been implemented in medicine to increase the immune system’s efficiency, sexual function, and antioxidative effects [19,20]. Previously, it was reported that ginseng has different protective effects on semen quality by enhancing sperm progressive motility in vitro [19]. Similarly, Kim and Hwang [21] reported that ginseng significantly increases sperm motility and membrane integrity of post-thaw sperm. Furthermore, ginseng is thought to stimulate the production of strong antioxidant activities in cryopreserved human semen [18,22]. Similarly, the result of ginseng treatment on enhancing motility and decreasing ROS was reported in boar semen cryopreservation [23]. Meanwhile, ginseng supplementation did not improve bull semen preservation [24]. 

To the best of our knowledge, the present study is the first to determine the effect of ginseng as an antioxidant supplement in freezing extenders on the quality of cryopreserved rooster semen. We determined optimal ginseng concentration and examined fertility.

## 2. Materials and Methods

### 2.1. Materials

The ginseng root from Panax quinquefolium (American ginseng) was purchased from Sigma-Aldrich (St. Louis, MO, USA) (product number: G7253), and the component of ginsenosides from American ginseng is documented (Rb1, Rb2, Re, Rc, Rg1, and Rd) [25]. Unless otherwise stated, all chemicals used in this study were purchased from Sigma-Aldrich (St. Louis, MO, USA).

### 2.2. Animals, Housing, and Feeding

Forty Thai native roosters (Pradu Hang Dum) aged one year were managed intensively in a battery cage of 60 × 45 × 45 cm dimensions for individual roosters; the lighting program consisted of natural daylight for approximately 12 h/day. All roosters received an estimated 130 g of commercial breeder feed (Balance 924, Betagro Company Limited, Nakhon Ratchasima, Thailand) that provided 17% protein, 3% fat, 6% fiber, and 13% moisture. Water was provided ad libitum throughout the study.

In 80 Thai native hens (Pradu Hang Dum) aged eight months, the lighting program consisted of natural daylight for approximately 12 h/day. All hens received an estimated 110 g of commercial breeder feed (U-Feed L3 ATM Farm Solution, APM Agro Company Limited, Ratchaburi, Thailand) that provided 18% protein, 3% fat, 6% fiber, and 13% moisture. Water was provided ad libitum throughout the study. 

### 2.3. Experimental Design and Ginseng Preparation

The optimal concentration of ginseng supplemented in the freezing extender for post-thaw rooster semen quality was determined. Based on a previous report on boar semen [23] and bull semen [24], the concentration of ginseng was examined at 0.25, 0.50, 0.75, and 1.00 mg/mL. The ginseng powder was weighed at 0,0.25, 0.50, 0.75, and 1.00 mg/mL, then was dissolved in a Schramm extender [26] by stirring for 1 min (using a vortex mixer) and incubated overnight at room temperature (28–30 °C) before being filtered with a sterile syringe filter of 0.22 µm to remove any particulates from the extender solution. The pooled rooster semen was subjected to one of the following five treatments depending on different doses of ginseng supplemented in a freezing extender [0 (control), 0.25, 0.50, 0.75, and 1.00 mg/mL]. After cryopreservation, the post-thaw semen was evaluated for sperm motility, membrane integrity, lipid peroxidation, and antioxidant enzyme activity. The best post-thaw semen quality was selected for further fertility tests. The experiment was replicated at least four times in each assessment.

### 2.4. Semen Samples Collection

Semen was routinely collected twice weekly for 12 weeks at 2.00–3.00 p.m. using the dorso-abdominal massage method [27]. The collection of semen samples from individual roosters was performed twice per week in a 1.5 mL microtube containing 100 µL of Schramm extender, composed of 0.7 g magnesium acetate, 28.5 g sodium glutamate, 5 g glucose, 2.5 g inositol, and 5 g potassium acetate, all dissolved in 1000 mL of deionized water; the pH was 7.1, and the osmotic pressure was 395 mOsm/kg. Total samples were preserved at an optimized temperature of 22–25 °C and transported to the laboratory for semen evaluation using conventional methods. 

### 2.5. Fresh Semen Evaluation

Semen volume was analyzed using a 1 mL syringe. The sperm concentration was determined using a hemocytometer under a light microscope at 400× magnification. One microliter of the semen sample was diluted with 999 µL of 4% sodium chloride. A loaded semen sample was entered into a hemocytometer, and counting was performed using a light microscope at 400× magnification. Sperm concentration was expressed as billion (10^9^) sperm cells/mL. Sperm motility was determined as mass motility (0–5 scales) and progressive motility under the light microscope at 100× and 400× magnification, respectively. About 5–10 µL of semen sample was dropped on the slide and examined immediately under a microscope without using a coverslip. Mass motility is scored on a scale of 0–5 (0 = no movement; 5 = very rapid waves and whirlpools visible). The progressive motility was evaluated with a light microscope at 400× magnification by diluting 5 µL of semen samples with 100 µL of 0.9% sodium chloride. Progressive motility was presented as a percentage of motile sperm. 

### 2.6. Dilution and Cryopreservation

Pooled semen was extended with a Schramm extender supplemented with different concentrations of ginseng according to the treatment groups at 1:3 (*v*:*v*) before gradually cooling down from 25 °C to 5 °C for about 45–60 min. Subsequently, 6% (*v*/*v*) of (N,N-dimethylformamide) was added to extended semen before loading semen into 0.5 mL plastic straws and cooling at 5 °C for 15 min. Then, they were cryopreserved using liquid nitrogen (LN_2_), according to the vapor method. All semen sample straws were laid horizontally on a rack 19 cm and 11 cm above the surface of LN_2_ (−35 °C and −135 °C) for 12 and 5 min, respectively. After that, they were plunged into LN_2_ for storage for about 2–4 weeks before thawing. The frozen semen straws were thawed at 5 °C for 5 min. They were maintained in cooled water during the sperm quality assessments.

### 2.7. Frozen-Thawed Semen Evaluation

#### 2.7.1. Sperm Motility

Total sperm motility and progressive motility were assessed using a computer-assisted sperm analysis system (Hamilton Thorne Biosciences, Beverly, MA, USA, version 12 TOX VIOS) equipped with Olympus software to process the video recorded. For analysis, 5 µL of frozen-thawed semen sample was loaded into a prewarmed (37 °C) analysis chamber. Image capture and motility settings were set up as follows: frame per second = 60; number of frames = 30; head brightness min = 165; minimum cell size = 2 μm; maximum cell size = 50. Evaluations of at least five fields with a minimum of 300 sperms per sample were performed. Total motility was expressed as the percentage of sperm exhibiting any movement. Progressive motility was expressed as sperm swimming in a straight line.

#### 2.7.2. Membrane Integrity

Sperm membrane integrity was evaluated using a fluorescent staining technique to separate live sperm with an intact plasma membrane and dead sperm using SYBR-14 and propidium iodide kits (L7011; Invitrogen, Thermo Fisher Scientific, Waltham, MA, USA). Briefly, 300 μL of the sperm suspension was mixed with 5 μL SYBR-14 solution and incubated at 24–27 °C for 10 min, then stained with 5 μL propidium iodide for 5 min. The samples were then fixed using 10% formaldehyde. Sperm were accessed, and at least 200 sperm cells were placed under an IX71 fluorescence microscope (Olympus, Tokyo, Japan) at 400× magnification and categorized into three groups as follows. Sperm with an intact plasma membrane was stained bright green with SYBR-14, whereas those with a damaged plasma membrane were stained red with propidium iodide. Some sperm cells, apparently stained with both red and green fluorescence, were classified as slightly damaged. Three major sperm populations are shown in Figure 1. Membrane integrity is expressed as the percentage of sperm with intact plasma membranes.

#### 2.7.3. Lipid Peroxidation

Lipid peroxidation was analyzed to determine MDA levels, the final product of lipid peroxidation. The lipid peroxidation reaction of sperm used thiobarbituric acid (TBA) as a test reagent to measure the amount of MDA. In the first assessment steps, 0.25 mL of ascorbic acid (1 mM) and 0.25 mL of ferrous sulfate (0.2 mM) were added to semen samples post-throwing. All substances were mixed and incubated at 37 °C for 60 min. To a sample of semen that has been finished from incubation, 1 mL of trichloroacetic acid [15% (*w*/*v*)] and 1 mL of TBA [0.375% (*w*/*v*)] were added. Next, samples were boiled at approximately 100 °C for 10 min. The sample was boiled for a specified time and placed in cold water at 4 °C, to stop the reaction between the reagent and the semen sample. Finally, the samples were separated from the supernatants (approximately 2 mL) via centrifuging at 800 rpm per 10 min, controlled at 4 °C. Finally, a semen sample was used to measure the amount of MDA using a UV-visible spectrophotometer at 532 nm (Analytik Jena Model Specord 250 Plus, Shimadzu Corporation, Kyoto, Japan).

#### 2.7.4. Enzyme Activity

The amount of antioxidant enzyme was determined by measuring the activities of SOD, CAT, and GPx using spectrophotometric analysis according to Nichi et al. [28].

##### SOD

A decrease in cytochrome C can be used to measure SOD activity because both SOD and cytochrome C compete with superoxide free radicals. Moreover, superoxide free radicals are converted into H_2_O_2_ and O^2−^. The decrease in cytochrome C results from the xanthine–xanthine oxidase system. For the analysis, 10 μL seminal plasma samples were mixed. The solution contained cytochrome C (1 mM), xanthine (50 mM), and 155 μL of xanthine oxidase diluted in sodium/EDTA buffer (50 and 100 mM, respectively, pH 7.8). The spectrophotometer was used to analyze the sample’s absorption every 5 min, and throughout the analysis, the samples were maintained at a temperature of 25 °C. Cytochrome C was calculated as a decreased rate of cytochrome C of 0.025 absorbance units/min (at a wavelength of 550 nm), and 1 unit of SOD accounted for 50% of this value. Therefore, SOD activity decreases cytochrome, C.

##### CAT

CAT activity was measured based on H_2_O_2_ quantity. Analysis was performed using 10 μL of seminal plasma and adding 90 μL of Tris (hydroxymethyl) amino methane/EDTA buffer solution (50 and 250 mM, respectively) and 900 μL of H_2_O_2_ (9.0 mM). Samples and solutions were chemically reacted at pH 8.0, 30 °C, for 8 min. The absorbance of the sample was measured at 230 nm using a spectrophotometer every 5 s. The resulting value was calculated as 0.071 M^−1^ cm^−1^, and the extinction coefficient for H_2_O_2_ CAT activity in seminal plasma was expressed as U/mL.

##### GPx

GPx activity was measured based on NADPH consumption. The reaction between H_2_O_2_ and reduced glutathione (GSH), catalyzed by GSPH-Px and the enzyme glutathione reductase (GSSGr), affected the conversion of glutathione disulfide to GSH, which in turn used NADPH. The solution used in the analysis consisted of NADPH (0.12 mM, 1 mL), GSH (1 mM, 100 mL), GSSGr (0.25 U/mL, 20 mL), and sodium azide (0.25 mM, 20 mL). Seminal plasma (100 mL) was then used. The spectrophotometer cell was brought to a volume of 1.9 mL with phosphate buffer 143 mM, EDTA 6.3 mM (pH 7.5), which was used to solubilize the NADPH. To prepare the GSH, 5% metaphosphoric acid was added. Sodium azide was used to inhibit CAT activity. The final initialization solution consisted of 100 mL of tert-butyl hydroperoxide 1.2 mM. The sample was measured in the absorption spectrometric wavelength of 340 nm for 10 min at 37 °C using a spectrophotometer, and the absorption value was measured every 5 s. The resulting value was calculated as follows: 6.22 mM^−1^ cm^−1^ as the extinction coefficient of NADPH; GSH-Px activity in seminal plasma was displayed as U/mL.

### 2.8. Fertility Test

The fertilizing test of the best post-thaw semen quality was selected for fertility tests compared with control by inseminating Thai native hens (Pradu Hang Dum) once a week with a dose of 0.4 mL of frozen-thawed semen with a final sperm concentration of 400 × 10^6^ sperm/hen from each group. Insemination was performed in the afternoon. Egg candling was used to examine fertility on Day 7 of incubation. Fertility was calculated as the percentage of fertile eggs in total eggs.

### 2.9. Statistical Analysis

The data were analyzed using a completely randomized design, and the treatment groups were compared using Tukey’s post hoc test. Before statistical analysis, the data were tested for normal distribution using the Shapiro–Wilk test and homogeneity of residual variances using Levene’s test, and outlier data were eliminated. The overall differences between the treatment means were considered statistically significant at *p* < 0.05.

## 3. Results

### 3.1. Fresh Semen Quality

The mean ± SE semen volume and sperm concentration of fresh semen samples were 400 ± 0.05 µL and 4.01 ± 0.25 × 10^9^ sperm/mL, respectively. The mean ± SE of mass and progressive motility were 4.15 ± 0.07 and 85.50 ± 0.90%, respectively. 

### 3.2. Frozen-Thawed Sperm Motility, Membrane Integrity, Lipid Peroxidation, and Enzyme Activities

The effects of ginseng supplementation in a freezing extender on the quality of frozen-thawed semen are shown in Table 1. Supplementation of a freezing extender with 0.25 mg/mL of ginseng resulted in the highest total motility, progressive motility, and membrane integrity (*p* < 0.05). Meanwhile, the supplementation of ginseng in the freezing extender at 0.75 and 1 mg/mL did not seem beneficial to sperm quality, as the results were similar to those of the control (*p* > 0.05).

The lipid peroxidation results, as indicated by MDA and the activities of the antioxidant enzymes SOD, CAT, and GPx, are shown in Figure 2. The highest MDA concentration was observed in the control group (*p* < 0.05). The ginseng supplementation in a freezing extender at 0.25 and 0.50 mg/mL resulted in higher CAT compared to 0.75 and 1.0 mg/mL groups (*p* < 0.05); meanwhile, the SOD and GPx levels among ginseng-supplemented groups were not different (*p* > 0.05). The lowest activity of antioxidant enzymes was observed in the control group. 

Based on the frozen-thawed sperm quality and lipid peroxidation results, ginseng supplemented in a freezing extender at 0.25 mg/mL was selected to study fertility.

### 3.3. Fertility Test

The effects of ginseng on fertility rates are presented in Figure 3. Fertility rates were compared between the control and 0.25 mg/mL of ginseng supplementation in a freezing extender. The results demonstrated that the percentages of fertility with 0.25 mg/mL of ginseng were higher than that of the control group (62.80 ± 4.20% vs. 46.28 ± 4.39%: *p* < 0.05).

## 4. Discussion

High quantities of polyunsaturated fatty acids in the sperm plasma membrane and low levels of antioxidant enzymes in the cytoplasm are the main reasons for sperm susceptibility to oxidative damage during cryopreservation [29,30]. To improve cryopreserved rooster semen quality, supplementing the freezing extender with several antioxidant substances is necessary to decrease the ROS levels occurring during the freezing process, leading to improved sperm membrane integrity after thawing, as reviewed by Partyka and Niżański [14]. The present study examined the effective dose of ginseng supplementation for freezing extenders of cryopreserved rooster semen. The 0.25 mg/mL ginseng dose improved frozen-thawed rooster semen quality, decreased lipid peroxidation, and increased enzyme activity after cryopreservation. Meanwhile, the supplementation of ginseng in a freezing extender at 0.75 and 1 mg/mL did not seem beneficial to sperm quality, as they were similar to those of the control. 

Ginseng has been demonstrated to have various effects, including antioxidative effects [31]. The predominant active ingredient of ginseng extract is ginsenosides which can be divided into two groups, protopanaxadiol- (PPD-) and protopanaxatriol- (PPT-) group ginsenosides [32]. The group of PPD-ginsenoside consists of Rb1, Rb2, Rb3, Rc, Rd, Rg3, and Rh2, while PPT-ginsenoside consists of Rg1, Rg2, and Rh1 [33]. The several components in the sample of this American ginseng root were found using HPLC’s separation of ginsenosides and have been reported as ginsenosides Rb1, Rb2, Re, Rc, Rg1, and Rd in the supplementary data. In this regard, Rb1, Rc, Re, and Rd seem to be the main components of American ginseng used in this study. From the review, the protective effect exerted by antioxidants can be explained by the ability of the significant substance of ginsenosides (Rb1, Rg1, and Rg2) to activate SOD, CAT, and GPx enzymes [33]. Ginsenoside Rc enhanced sperm motility in human semen samples [34]; meanwhile, ginsenoside Re stimulated nitric oxide production by inducing nitric oxide synthase activity; promoted sperm capacitation and acrosome reaction; and enhanced the fertilizing capacity in humans [35]. The ginsenoside Rd was reported to have an efficacy capacity to decrease the reactive oxygen species in cultured cells [36]. Therefore, it could be assumed that American ginseng used in this study has various antioxidant activities and positively affects rooster sperm. We confirm its effect by determining the generation of ROS and the concentrations of antioxidant enzymes in cryopreserved semen. The results of this study demonstrated that treatment with ginseng in a freezing extender decreased MDA and increased enzyme activity; meanwhile, the highest MDA and the lowest total activity of antioxidant enzymes was found in the control (Figure 2). Hu and Kitts [37] reported that antioxidant enzymes play an important role in inhibiting the generation of ROS by scavenging free radicals; thus, the sperm plasma membranes were preserved. Therefore, sperm motility and membrane integrity were improved after supplementation of ginseng in a freezing extender. 

Considering the effective dose among species, we noticed that the effective amount of ginseng in rooster sperm was 0.25 mg/mL, while that in boar was 0.75 mg/mL. It might be inferred that the difference in antioxidant supplement doses in each species followed their difference in cholesterol to phospholipids (C/PL) ratio among species [38]. A high C/PL ratio allows the membrane to maintain fluidity during freezing. Therefore, a lower C/PL ratio in boar (0.26) required more ginseng than for the higher ratio in roosters (0.30). However, the antioxidant activity of ginseng in cryopreserved semen was not dose-dependent. The present study found that supplementing ginseng in a freezing extender (0.75 and 1 mg/mL) did not seem beneficial for sperm quality, which was consistent with previous studies that determined that high levels of ginseng supplementation between 1.0 and 7.5 mg/mL in chilled semen and cryopreservation of bull semen hurt semen quality [24]. Increasing antioxidant concentrations may result in hypertonic conditions, causing the dehydration of sperm cells [39]. In addition, many studies have reported the high antioxidant toxicity of semen extenders. For instance, Vongpralub et al. [40] suggested that high doses of vitamin E (1 and 5 mM) were toxic to boar sperm. In addition, the study on sericin in boar and bull cryopreserved semen results showed that a higher sericin level of 1.0% in the freezing extender negatively affected the sperm quality [41,42]; therefore, the optimal level in each species is not similar and important to the success of semen cryopreservation.

In addition to improved post-thaw semen quality, supplementation with 0.25 mg/mL of ginseng in the freezing extender improved fertility compared with the control group (Figure 3). It is suggested that a significant increase in sperm membrane integrity and motility after freezing increased the fertility of native hens to achieve better results. This result is consistent with that of human studies showing that red ginseng improves male reproductive function and infertility [43]. Fertility success using semen freezing in poultry varies and depends on the species and different antioxidants to protect sperm cells. Many studies of rooster semen have improved semen quality after being frozen-thawed by supplementing antioxidants in the diluent as the natural antioxidants in the seminal plasma; for instance, vitamin A, C, E, and glutathione are enough to stabilize sperm cell membranes [14] but insufficient to protect sperm cells from excessive ROS during cryopreservation [15]. A significant decrease in damage to the cryopreserved sperm plasma membrane or maintaining sperm membrane integrity is often reported in increasing fertility. The researchers obtained fertility of approximately 60%, either for the motility or viability of frozen-thawed semen up to 60%, by supplementing antioxidants in a freezing extender such as glutathione [44], coenzyme Q10 [45], and resveratrol and resveratrol-loaded nanostructured lipid carriers [46]. In a word, several effective semen freezing additives such as antioxidants are being developed for rooster sperm preservation. However, different factors such as breed, age, and environmental temperatures might influence the antioxidant status and should be considered.

## 5. Conclusions

In conclusion, our findings indicate that supplementing 0.25 mg/mL of ginseng is recommended as an antioxidant in the freezing extender for improving frozen-thawed semen quality, reducing oxidative stress during the freezing process, enhancing enzyme activity in seminal plasma, and increasing fertility in rooster semen cryopreservation. 

## Figures and Tables

**Figure 1 animals-13-02660-f001:**
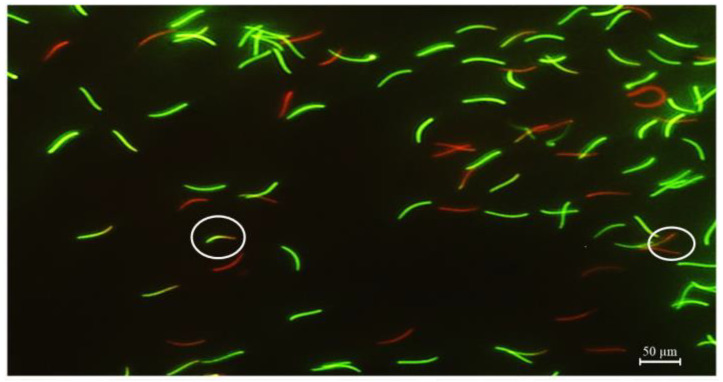
Rooster sperm were fluorescently stained using a combination of SYBR-14 and propidium iodide at 400× magnification. Three major sperm populations were classified. The sperm with an intact membrane stained bright green (SYBR-14); the sperm with damaged membranes stained red (PI); and the slightly damaged sperm stained both red and green fluorescence (circle).

**Figure 2 animals-13-02660-f002:**
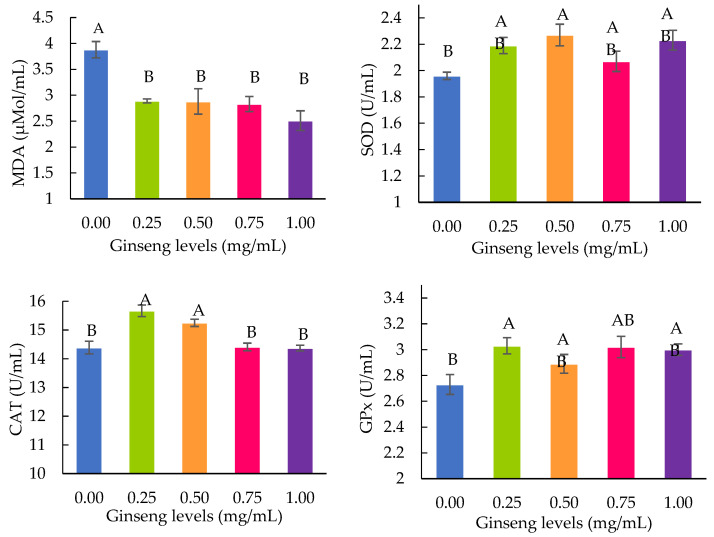
Effects of different concentrations of ginseng on lipid peroxidation and enzyme activity (SOD = superoxide dismutase; CAT = catalase; and GPx = glutathione peroxidase) of frozen-thawed rooster semen. Within parameters, mean ± standard error with different alphabets differed significantly (*p* < 0.05). MDA = malondialdehyde.

**Figure 3 animals-13-02660-f003:**
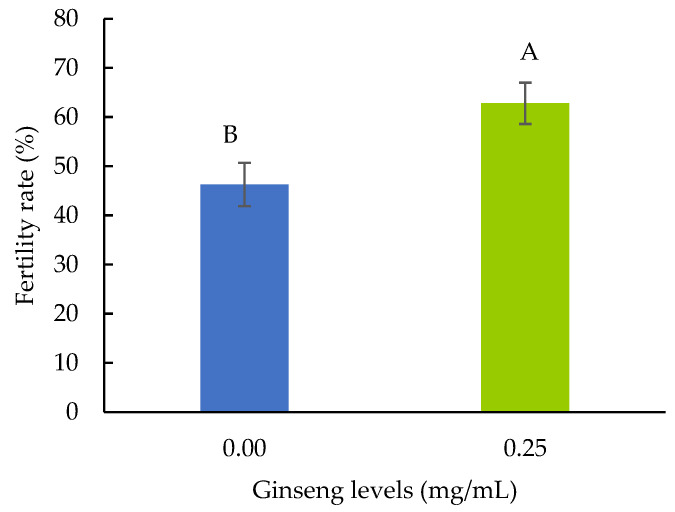
Effects of ginseng on fertility rate. Within parameters, mean ± standard error with different alphabets differed significantly (*p* < 0.05).

**Table 1 animals-13-02660-t001:** Effects of different concentrations of ginseng on total motility, progressive motility, and membrane integrity in frozen-thawed semen (mean ± SE).

Ginseng Levels (mg/mL)	Parameters		
	MOT (%)	PMOT (%)	Membrane Integrity (%)
0	38.18 ± 0.56 ^BC^	24.88 ± 0.38 ^B^	26.14 ± 3.98 ^C^
0.25	59.47 ± 2.26 ^A^	30.82 ± 0.70 ^A^	48.30 ± 1.70 ^A^
0.50	46.22 ± 1.42 ^AB^	26.80 ± 2.10 ^AB^	35.71 ± 2.29 ^B^
0.75	38.47 ± 3.69 ^BC^	24.75 ± 0.90 ^B^	27.15 ± 0.88 ^BC^
1	30.67 ± 3.07 ^C^	25.25 ± 0.68 ^AB^	23.90 ± 1.48 ^C^
*p*-value	0.0002	0.0274	<0.0001

Different letters (^A,B,C^) within columns indicate significant differences (*p* < 0.05). MOT = total motility; PMOT = progressive motility.

## Data Availability

The data are available upon request from the corresponding author.

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
