# Peer review of "Supplementation of Freezing Medium with Ginseng Improves Rooster Sperm Quality and Fertility Relative to Free Radicals and Antioxidant Enzymes"

_animals, 2023, doi:10.3390/ani13162660_

Round 1
Reviewer 1 Report
This is a very interesting and complete study and of great interest to the poultry industry. The authors, working with a relatively high number of frozen rooster semen samples, manage to demonstrate, through a complete battery of in vitro tests, than the use of an antioxidant, such as ginseng at a concentration of 0.25 mg/mL improve rooster semen cryopreservation. But, the number of experiment results in this article is not enough for fundamental research regarding spermatozoa, it is recommended author to add more wet-lab results. After a major revision, the manuscript would probably be accepted.
Figures with nice pictures would be more readable for a scientific publication. Figure 1-2: adjust the color scheme to visualize the parameters. The authors should add more original photographs from microscopy or flow cytometry. The author can read more examples from the following articles.
[1] Stress decreases spermatozoa quality and induces molecular alterations in zebrafish progeny. BMC biology, 2023.
[2] Influence of Two Widely Used Solvents, Ethanol and Dimethyl Sulfoxide, on Human Sperm Parameters. 2022.
[3] Effect of Zinc on Boar Sperm Liquid Storage. 2023.
[4] Synergistic Effects of Myo-Inositol and Melatonin on Cryopreservation of Goat Spermatozoa. Reprod Domest Anim, 2022.
[5] Effect of Mitochondria-Targeted Antioxidant on the Regulation of the Mitochondrial Function of Sperm During Cryopreservation. Andrologia, 2022.
[6] Cryo-banking of human spermatozoa by aseptic cryoprotectants-free vitrification in liquid air: Positive effect of elevated warming temperature. Cell and tissue banking, 2022.
Please carefully check all the sentences, and make sure there is a space between each sentence.
The third paragraph of the introductory section has too much subject matter.
It is recommended to rewrite the full text of Sperm as Spermatozoa
Detailed Comments:
The Materials and Methods are not sufficiently detailed to allow the reader to replicate the experiments.
The author should briefly introduce the dietary formulas of roosters and hens.
The extender should include references
The authors should expand their discussion
Reviewer 2 Report
The manuscript 'Supplementation of freezing medium with ginseng improves rooster sperm quality and fertility relative to free radicals and antioxidant enzymes' has been reviewed thoroughly. The study on the effect of Ginseng root extract on sperm quality and fertility has been carried out in rooster and found to improve the sperm quality and fertility after cryopreservation. In general, the manuscript was written well and it is an interesting study; however, there exist some flaws in the experiment design and some places need to be addressed.
Majors:
1. How did the authors prepare the Ginseng root extract?
2. What is the content of total ginsenosides in the extract?
3. What are the main components of ginsenosides in the extract used?
4. Section 2.2: what the authors stated that“An extender without ginseng root extract served as the control”is not correct. The authors should use the extender containing the same volume of the diluent buffer as the control, which was used for the dilution of the extract at different concentrations.
5. Since the rooster sperm cryopreservation caused membrane and sperm damage and the authors thought that the Ginseng root extract could protect sperm membranes against peroxidative damage, the authors should show the data about any change of the membranes of the sperm treated with the Ginseng root extract or vehicle during the cryopreservation.
Minors:
1. Figure 1: The title of the X-axis should be “Content of Ginseng root extract (mg/ml)” instead of “Ginseng level (mg)”.
2. Figure 1 again: It is not self-evidence enough to read easily. In addition, the unit (mg) after the number (0 and 0.25) below the horizontal axis should be deleted; it should be expressed as “0” and “0.25” instead of “0 mg” and “0.25 mg”, respectively.
Reviewer 3 Report
This is an interesting manuscript related to the use of ginseng extract as a supplement to avoid oxidative stress during rooster semen cryopreservation. Manuscript is well written in general, but some points should be adjusted as listed below:
1. Abstract should present some numeric results, specially related to the sperm parameters that presented significant differences as sperm motility.
2. In the first paragraph of introduction, authors highlight that "Sperm cryopreservation is an emerging potential tool for conserving the genetic re-sources of indigenous species that have experienced a tremendous decline in their numbers and are considered endangered." In my point of vie, authors should now present the avian species that they will use at the study and highlight its status of conservation. On the contrary, if it is not endangered, they should highlight its use as experimental model for the endangered ones.
2. In material and methods, authors should highlight the origin of the ginseng extract concentrations used at the study. Were they previously used for other species?
3. Also, authors should provide references for semen collection method, even if it is largely used.
4. Authors should indicate the average time that samples were kept stored in liquid nitrogen before thawing.
5. Regarding sperm motility for frozen thawed semen, authors should provide the settings used for CASA, and report the chamber used for the analysis. Also, they should indicate the parameters evaluated.
6. For CASA and other analysis, please provide references.
7. In fertility test, were semen samples from different males pooled? or were they individually used? If yes, what was the proportion of females inseminated with the semen from each male?
8. In results section, please present the average values for fresh samples.
9. Why only sperm total and progressive motility are presented for frozen-thawed samples? Were are the other CASA parameters?
10. Please include some discussions related to the practicality of using ginseng extract. Highlight its advantages and list the possible disadvantages. Also, try to report a conceptual comparison to the use of other antioxidants previously used for rooster frozen-thawed sperm.
Round 2
Reviewer 1 Report
Accept in present form
Author Response
We are grateful for your opinion that believing in the information and its worth in publishing.
Reviewer 2 Report
The manuscript 'Supplementation of freezing medium with ginseng improves rooster sperm quality and fertility relative to free radicals and antioxidant enzymes' has been revised according to the reviewers’ suggestions. The authors modified the description of the key material “Ginseng”. In the revised version, the authors called the material “Ginseng” instead of “Ginseng root extract”; it is a very important revision! If the “Ginseng root extract” was used in the experiment, the detailed procedure of the extract preparation should be given in the Materials and methods section. However, some other issues are needed to be addressed.
Majors:
1. the authors should describe the preparation of the “Ginseng root” solution in the section Materials and Methods. How did the authors make the chemical “Ginseng root” a solution that could be used for treatment of the sperm? Actually, how to prepare the Ginseng root solution? Was the “root” dissolved easily in water?
2. What is the content of total ginsenosides in the “Ginseng root” that was obtained from Sigma-Aldrich?
3. What are the main components of ginsenosides in the Ginseng root used since the authors discussed the effective active ingredients of Ginseng? The authors should give the information of the active ingredients in the Ginseng root used.
4. Section 2.1: “the component of their concentration is ≤ 100 percent”is not clear. What is the component of their concentration?
5. Since the rooster sperm cryopreservation caused membrane and sperm damage and the authors thought that the Ginseng could protect sperm membranes against peroxidative damage, the authors should show the data about any change of the membranes of the sperm treated with the Ginseng root solution or vehicle during the cryopreservation.
Minors:
1. Abstract and section 3.3: “62.80±4.20% and 46.28±4.39%, respectively (p<0.05)”should be changed to “62.80±4.20% vs 46.28±4.39%)”.
2. The number of the reference [27] cited in the section 2.4 should be moved to section 2.3 and cited after“Schramm extender”.
It is OK.
